# Predictors of mortality in treatment experienced HIV-infected patients in northern Tanzania

Deng B. Madut[1]*, Lawrence P. Park[1,2], Jia Yao[3,4], Elizabeth A. Reddy[5], Bernard Njau[6], Jan Ostermann[7], Kathryn Whetten[2,3,4], Nathan M. Thielman[1,2]

1 Department of Medicine, Duke University, Durham, NC, United States of America, 2 Duke Global Health Institute, Durham, NC, United States of America, 3 Center for Health Policy and Inequalities Research, Durham, NC, United States of America, 4 Duke Sanford School of Public Policy, Duke University, Durham, NC, United States of America, 5 Division of Infectious Disease, SUNY Upstate Medical University, Syracuse, NY, United States of America, 6 Kilimanjaro Christian Medical University College, Moshi, Tanzania, 7 Department of Health Services Policy and Management, Arnold School of Public Health, University of South Carolina, Columbia, South Carolina, United States of America

* deng.madut@duke.edu

## Abstract

### Background

While factors that drive early mortality among people living with HIV (PLWH) initiating antiretroviral therapy (ART) in sub-Saharan Africa (SSA) have been described, less is known about the predictors of long-term mortality for those with ART experience.

### Methods

PLWH and on ART attending two HIV treatment clinics in Moshi, Tanzania were enrolled from 2008 through 2009 and followed for 3.5 years. Demographic, psychosocial, and clinical information were collected at enrollment. Plasma HIV RNA measurements were collected annually. Cause of death was adjudicated by two independent reviewers based on verbal autopsy information and medical records. Bivariable and multivariable analyses were conducted using Cox proportional hazard models to identify predictors of mortality.

### Results

The analysis included 403 participants. The median (IQR) age in years was 42 (36–48) and 277 (68.7%) participants were female. The proportion of participants virologically suppressed during the 4 collection time points was 88.5%, 94.7%, 91.5%, and 94.5%. During follow-up, 24 participants died; the overall mortality rate was 1.8 deaths per 100 person-years. Of the deaths, 14 (58.3%) were suspected to be HIV/AIDS related. Predictors of mortality (adjusted hazard ratio, 95% confidence interval) were male sex (2.63, 1.01–6.83), secondary or higher education (7.70, 3.02–19.60), receiving care at the regional referral hospital in comparison to the larger zonal referral hospital (6.33, 1.93–20.76), and moderate to severe depression symptoms (6.35, 1.69–23.87).

**Data Availability Statement:** Data cannot be publicly shared because it contains potential identifiers and is considered sensitive data. Data will be made available for researchers who can

meet the criteria for accessing sensitive and confidential data, including obtaining ethics committee review and complying with data management and transfer agreements. Requests may be sent to Kimberly McNeil, Associate Director of the Center for Health Policy & Inequalities Research at Duke University (km.mcneil@duke.edu).

**Funding:** The CHAT study was funded by Grant 5R01MH078756 from the National Institute of Mental Health, with additional support from the Duke Center for AIDS Research, 5P30-AI064518; the Duke Interdisciplinary Research Training Program AIDS, T32 AI007392; the Vanderbilt-Emory-Cornell Duke Consortium for Global Health, D43TW009337; and the Hubert-Yeargan Center for Global Health, Duke University. DBM received support from the NIH Ruth L. Kirschstein National Research Service Award (NRSA) T32AI007392 and US NIH Fogarty International Center grant D43TW009337. The funders had no role in study design, data collection and analysis, decision to publish, or preparation of the manuscript.

**Competing interests:** The authors have declared that no competing interests exist.

## Conclusions

As ART coverage continues to expand in SSA, HIV programs should recognize the need for interventions to promote HIV care engagement for men and the integration of mental health screening and treatment with HIV care. Facility-level barriers may contribute to challenges faced by PLWH as they progress through the HIV care continuum, and further understanding of these barriers is needed. The association of higher educational attainment with mortality merits further investigation.

## Introduction

Mortality among persons living with HIV (PLWH) in sub-Saharan Africa (SSA) has declined remarkably since the rapid scale-up of antiretroviral therapy (ART) [1]. Despite this progress, mortality rates for PLWH on ART in SSA remains high in comparison to high-resource settings [2]. In order to optimize the effectiveness of ART, especially in the context of expanding ART coverage, there is a need to understand factors associated with mortality among PLWH on ART.

Much of the excess mortality among PLWH on ART in SSA is the consequence of delayed ART initiation [3]. ART initiation at advanced stages of HIV disease translates into high early mortality, with the majority of deaths occurring within the first few months, often secondary to opportunistic infections [4]. Previous studies indicate that mortality rates decrease for those who survive the initial 12–24 months after ART initiation [5, 6]. Despite this, mortality rates among PLWH remains higher than that of background populations even in the setting of immune recovery and ongoing virologic suppression [7, 8]. Furthermore, prognostic factors predictive of early mortality (for example CD4 count at ART initiation) have diminishing prognostic value with increasing treatment experience [5, 9]. Given the increasing number of PLWH currently receiving ART in SSA, knowledge of the factors that influence long-term mortality could lead to further optimization of HIV programs.

The objective of this study is to examine the factors associated with mortality among a cohort of treatment-experienced HIV-infected individuals enrolled in the Coping with HIV/AIDS in Tanzania (CHAT) study. We further explore this cohort by describing virologic suppression and causes of death.

## Methods

### Ethical considerations

Study activities were approved by the Kilimanjaro Christian Medical Center Institutional Review Board (IRB) in Tanzania and the Duke University Health System IRB in the USA. Written informed consent was obtained from all participants.

### Participants

The Coping with HIV/AIDS in Tanzania (CHAT) study was an observational cohort study designed to examine the relationships between psychosocial characteristics, HIV medication adherence, and health outcomes among PLWH in Tanzania [10–16]. The study recruited 1,197 participants in Moshi, Tanzania from November 2008 through October 2009 into four distinct cohorts: a clinical cohort of patients with established HIV infection receiving care at two HIV clinics, individuals having just tested HIV-positive at voluntary counseling and testing

(VCT) sites; individuals having just tested HIV-negative at a VCT site; and a random sample of adults from the surrounding community. This analysis was restricted to the clinical cohort.

Participants in the clinical cohort were receiving care at the HIV clinics of Kilimanjaro Christian Medical Centre (KCMC) or Mawenzi Regional Referral Hospital (MRRH). KCMC, henceforth referred to as the "private zonal referral hospital", is a 630-bed private zonal referral hospital that serves the regions of Kilimanjaro, Tanga, Arusha, Manyara, Dodoma, and Singida in the northern zone of Tanzania (Fig 1). MRRH, henceforth referred to as "public regional hospital", is a 300-bed public regional hospital serving the Kilimanjaro Region.

Specific recruitment procedures for the clinical cohort were as follows. Any patient 18–65 years old and residing in the districts of Moshi Urban, Moshi Rural, Hai of the Kilimanjaro Region, and with plans to stay in the region for the foreseeable future, was eligible for enrollment. Due to staffing limitations and the interview length, a maximum of 3 participants per clinic day could be enrolled at each clinic. At both clinics, it was estimated that at least 30% of patient visits were unscheduled, and there was a wide variation in the number of patients presenting on any given clinic day. As previously described [17], participants were selected by a random time point system. In order to construct the selection parameters for the random time selection, clinic flow was observed over the period of one week. Based on this observation,

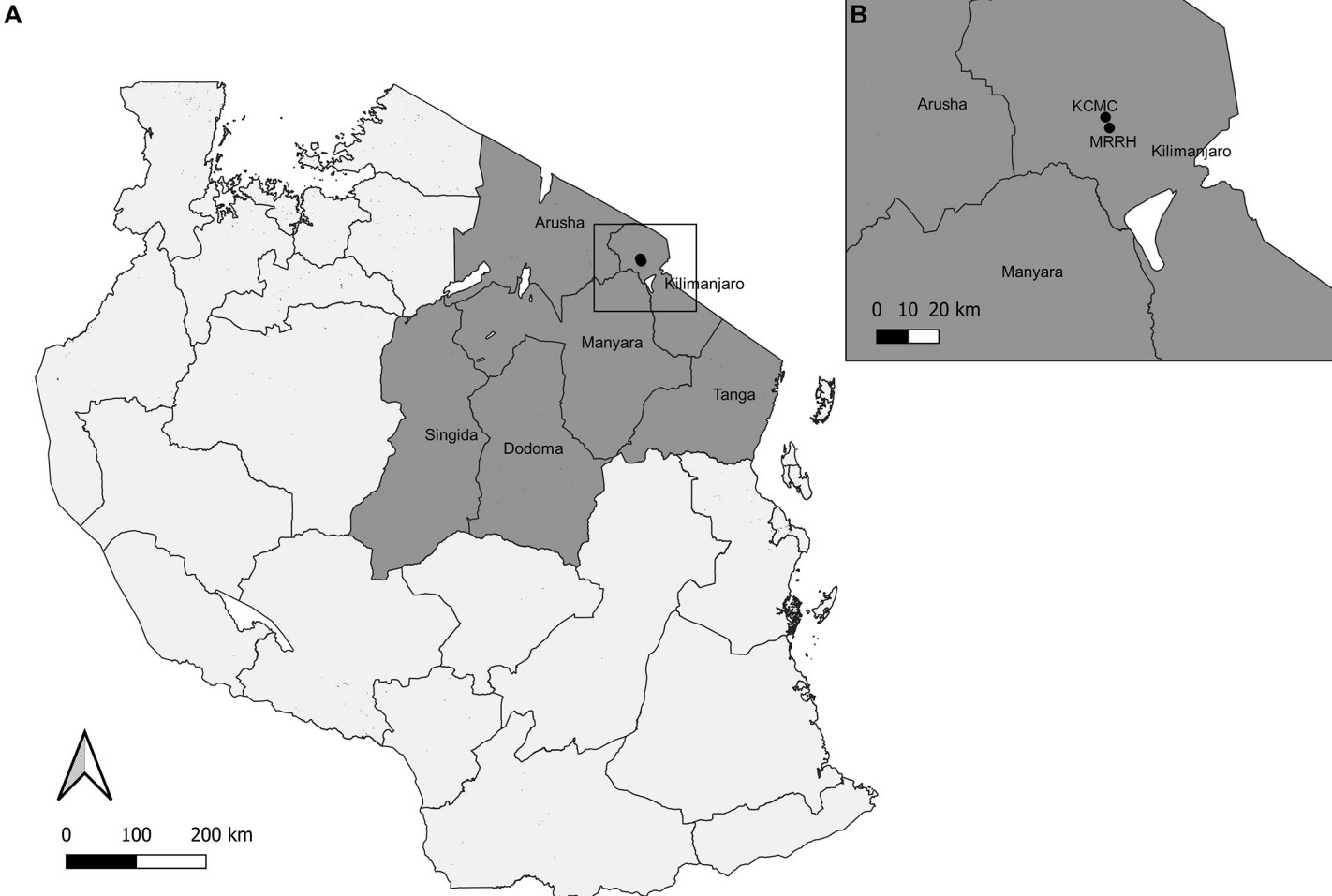

**Fig 1.** Map of Tanzania sub-divided into regions (A) Regions of Kilimanjaro, Tanga, Arusha, Manyara, Dodoma, and Singida are labeled and Mawenzi Regional Referral Hospital (MRRH) and Kilimanjaro Christian Medical Centre (KCMC) are identified by point markers. (B) Point markers for MRRH and KCMC.

three minutes from the clinic day were randomly selected, with the probability of selection proportional to the expected number of patients in a given time interval. The selected minutes for each day were distributed to the clinic nurses and programmed into alarm clocks placed at the nursing triage station. After an alarm went off, the next patient in line for triage was screened for eligibility and, if eligible, read a standardized brief description of the research. Patients who were interested and desired further information underwent the full consent process with a research staff. If the selected patient declined or was ineligible, the next patient was approached as a replacement.

## Measures

**Demographic characteristics.** Participants completed in-person interviews every six months for 3.5 years, resulting in eight rounds of interviews. All interviews were conducted in Kiswahili. Demographic information collected at baseline included age, sex, highest level of education attained, and marital status.

**Clinical characteristics.** Depressive symptoms were measured using the Patient Health Questionnaire (PHQ-9), a nine-item questionnaire which has been validated in African populations [18, 19]. Established categories for the PHQ-9 are no depression (sum score 0–4), mild depression (5–9), moderate depression (10–14), moderately severe depression (15–19), and severe depression (20–27) [20]. We dichotomized participants as exhibiting no to mild depression (0–9) and moderate to severe depression ($\geq$10). ART adherence was assessed with two self-report questions. First, each participant was asked when they last missed a dose of their ART, with response options of never, within the past week, 1–2 weeks ago, 3–4 weeks ago, 1–3 months ago, or more than 3 months ago. Next, the participant was asked to show, on a 0–100% visual analog scale, the percentage of their ART they had taken in the last month and the percent of their ART they had missed in the last month. Participants were classified as adherent if they reported not missing doses. CD4 counts were obtained through abstraction from medical records. The nadir CD4 count was the lowest CD4 count recorded prior to enrollment. Plasma HIV RNA levels were measured at baseline and annually thereafter for a total of four collections during the duration of the study. For this analysis, virologic suppression is defined as HIV RNA level less than 1000 copies per mL for an individual taking ART for at least 6 months. All predictor variables were taken from the baseline survey.

**Loss to follow-up.** All efforts were made to minimize loss to follow-up. For participants not returning to the HIV clinic within 4 weeks of a scheduled follow-up assessment, home-based visits were conducted to ascertain the reasons, to collect all relevant information, and to invite the participant to return to the clinic for the scheduled clinical assessment. Participants lost to follow-up remained eligible for follow-up visits until the end of data collection as long as they fulfilled the initial enrollment criteria.

**Mortality ascertainment.** In the event of death, interviewers used a verbal autopsy survey to interview designated family and friends of the deceased participant from whom consent had been obtained. The verbal autopsy tool was adapted from the World Health Organization Verbal Autopsy Standard [21]. Additional information, if available, was abstracted from medical charts at affiliated clinics, death certificates, and inpatient documents. Two physicians (NT and ER) independently reviewed the available data and assigned the causes of death. They then determined if each cause of death was secondary to the underlying diagnosis of HIV/AIDS. When there was initial disagreement, data were reviewed together, and a final adjudication decision was made. All deaths for which agreements could be made between the two reviewers were classified as "likely HIV/AIDS related", "possibly HIV/AIDS related", or "not clearly HIV/AIDS related". If an agreement was not reached, the case was classified as "other".

## Analysis

Continuous variables are described using the mean and standard deviation (SD) or the median and interquartile range (IQR). Categorial variables are described using frequencies. Days of follow-up were calculated between the date of enrollment and the date of death, or, for non-decedents, right-censoring at the time of their last follow-up interview. Kaplan-Meier curves were used to estimate survival after enrollment. Bivariable Cox proportional hazards models were used to determine the relationship between independent predictor variables and mortality. In multivariable analysis, we used a block-wise selection process with *a priori* selected variables based on theory and prior research. Model 1 included relevant sociodemographic variables. Model 2 included age, sex, and clinically relevant variables. Model 3 included all variables. The proportional hazards assumption was tested using scaled Schoenfeld residuals. All data analysis was performed using STATA version 15.0 (StataCorp, College Station, TX) and $P$ values $<0.05$ were considered statistically significant.

## Results

### Participant characteristics

From November 2008 through October 2009, we enrolled 499 PLWH at the HIV clinics of the private zonal referral hospital and the public regional hospital (Fig 2). Of these 499

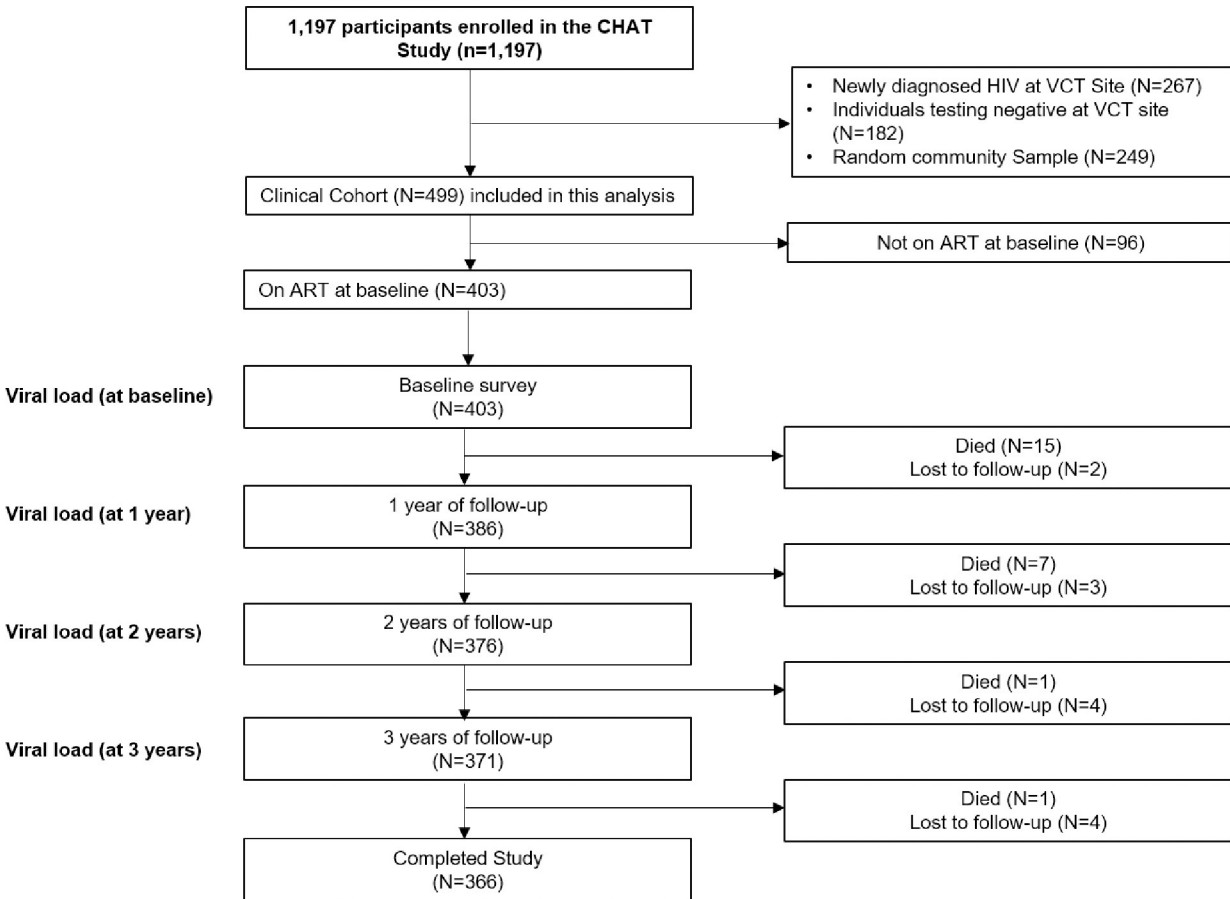

**Fig 2. Flow diagram of study participants enrolled in the coping with HIV/AIDS in Tanzania (CHAT) study, northern Tanzania (2008–2012).**

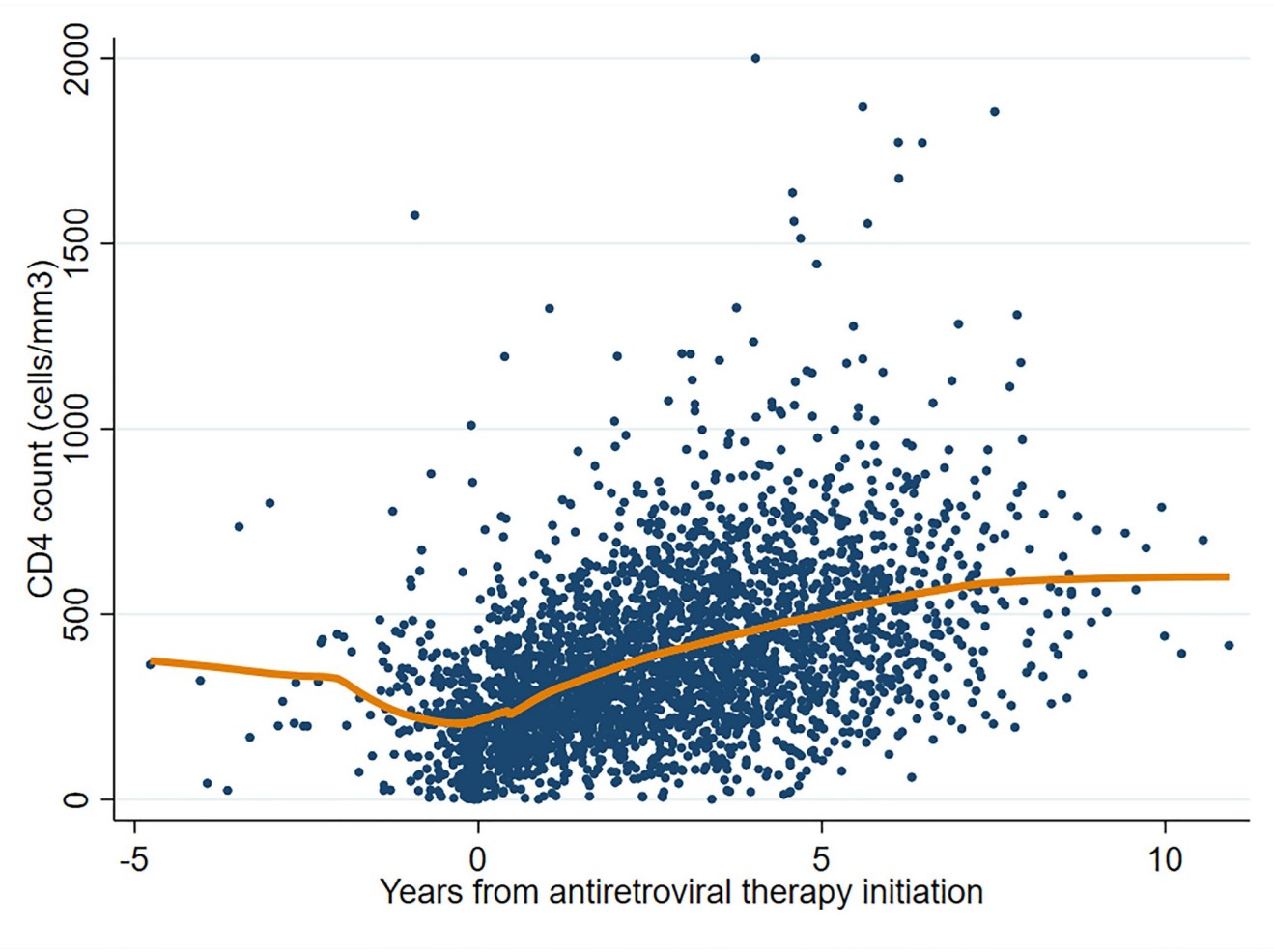

**Fig 3. CD4 count trends from time of ART initiation among a cohort of HIV-infected individuals on ART in Tanzania.**

participants, 403 were on ART at the time of enrollment and thus included in this analysis. The mean (SD) age in years of the analyzed cohort was 42 (8.3) and 277 (68.7%) were female. At enrollment, the median (IQR) days on ART was 703 (306–1187). The median (IQR) nadir CD4 count was 108 (45–180), but CD4 counts recovered after ART initiation (Fig 3). The proportion of patients virologically suppressed during each of the 4 collection time points was 88.5%, 94.7%, 91.5%, and 94.5% respectively. Participant characteristics are presented in Table 1.

Of the 403 participants analyzed, 24 died during 1,348 person-years of follow-up. The mortality rate was 1.8 deaths per 100 person-years. The median follow-up time for survivors was 3.5 years and the median follow-up time for decedents was 1.3 years. Of the 24 deaths, 9 (37.5%) were suspected to be likely HIV/AIDS related, 5 (20.8%) possibly HIV/AIDS related, and 2 (8.3%) were not HIV/AIDS related. An agreement regarding the cause of death could not be reached for 8 (33.3%) deaths.

## Predictors of mortality

In multivariable analysis (Table 2), significant predictors of mortality (adjusted hazard ratio [aHR], 95% confidence interval) were male sex (aHR 2.63, 1.01–6.83), attainment of secondary or higher education (aHR 7.70, 3.02–19.60), receiving treatment at a public regional referral

**Table 1. Distribution of demographic characteristics, clinical and laboratory findings by mortality status among a cohort of HIV-infected patients on antiretroviral therapy in northern Tanzania, 2008–2012.**

| Risk factors | Entire cohort (n = 403) | Survivors (n = 379) | Decedents (n = 24) |
|---|---|---|---|
| Age in years, mean (SD) | 42 (8.3) | 42.3 (8.3) | 43.0 (9.0) |
| Age in years, n (%) | | | |
| < 35 years | 65 (16.1) | 62 (16.4) | 3 (12.5) |
| ≥ 35 to < 45 years | 184 (45.7) | 173 (45.7) | 11 (45.8) |
| ≥ 45 years | 154 (38.2) | 144 (37.9) | 10 (41.7) |
| Sex, n (%) | | | |
| Female | 277 (68.7) | 267 (70.5) | 10 (41.7) |
| Male | 126 (31.3) | 112 (29.5) | 14 (58.3) |
| Education level, n (%) | | | |
| Primary or less | 313 (78.4) | 301 (80.7) | 12 (50.0) |
| Secondary or higher | 84 (21.1) | 72 (19.3) | 12 (50.0) |
| Missing | 6 | 6 | 0 |
| Marital Status, n (%) | | | |
| Married or cohabitating | 149 (37.5) | 141 (37.8) | 8 (33.3) |
| Divorced | 81 (20.4) | 73 (19.6) | 8 (33.3) |
| Widowed | 108 (27.2) | 104 (27.9) | 4 (16.7) |
| Never Married | 59 (14.9) | 55 (14.8) | 4 (16.7) |
| Missing | 6 | 6 | 0 |
| Clinic location, n (%) | | | |
| Private zonal referral hospital | 196 (48.6) | 190 (50.1) | 6 (25.0) |
| Public regional hospital | 207 (51.4) | 189 (49.9) | 18 (75.0) |
| Depressive Symptoms, n (%) | | | |
| No to mild | 358 (90.4) | 339 (90.9) | 19 (82.6) |
| Moderate to severe | 28 (9.6) | 34 (9.1) | 4 (17.4) |
| Missing data | 7 | 6 | 1 |
| ART Adherence, n (%) | | | |
| Complete adherence | 352 (91.7) | 332 (91.7) | 20 (90.9) |
| Incomplete adherence | 32 (8.3) | 30 (8.3) | 2 (9.1) |
| Missing data | 19 | 17 | 2 |
| CD4 nadir, median (IQR) cells/mm$^3$ | 108 (45–180) | 118 (50–190) | 60 (37–144) |
| CD4 nadir, n (%) | | | |
| < 200 cells/mm$^3$ | 315 (78.2) | 293 (77.3) | 22 (91.7) |
| ≥ 200 cells/mm$^3$ | 88 (21.8) | 86 (22.7) | 2 (8.3) |
| Time on ART, median (IQR) days | 703 (306–1187) | 698 (306–1190) | 729 (233–1116) |
| Time on ART at enrollment, n (%) | | | |
| < 1 year | 115 (28.5) | 108 (28.5) | 7 (29.2) |
| ≥ 1 year | 288 (71.5) | 271 (71.5) | 17 (70.8) |
| Viral suppression (at baseline), n (%) | | | |
| Suppressed | 322 (88.5) | 306 (89.5) | 16 (72.7) |
| Missing data | 39 | 37 | 2 |
| Viral suppression (at 1 year), n (%) | | | |
| Suppressed | 326 (94.2) | 319 (94.7) | 7 (77.8) |
| Missing data | 42 | 40 | 2 |
| Viral suppression (at 2 year), n (%) | | | |
| Suppressed | 321 (91.5) | 320 (91.7) | 1 (50.0) |
| Missing data | 26 | 25 | 1 |

*(Continued)*

**Table 1.** (Continued)

| Risk factors | Entire cohort (n = 403) | Survivors (n = 379) | Decedents (n = 24) |
|---|---|---|---|
| Viral suppression (at 3 year), n (%) | | | |
| Suppressed | 325 (94.2) | 324 (94.2) | 1 (100.0) |
| Missing data | 26 | 26 | 0 |

Abbreviations: ART, antiretroviral therapy.

hospital (aHR 6.33, 1.93–20.76), and experiencing moderate to severe depression symptoms versus no to mild depression symptoms (aHR 6.35, 1.69–23.87).

## Discussion

We found a mortality rate of 1.8 deaths per 100 person-years among PLWH receiving ART in northern Tanzania. Male sex, attainment of secondary education or higher, receiving care at the regional referral hospital (in comparison to the larger zonal referral hospital), and

**Table 2. Multivariable analysis of risk factors for death among a cohort of HIV-infected patients on antiretroviral therapy in northern Tanzania, 2008–2012.**

| Risk factors | Unadjusted HR (95% CI) | Model 1 aHR (95% CI) | Model 2 aHR (95% CI) | Model 3 aHR (95% CI) |
|---|---|---|---|---|
| Age in years | 1.01 (0.96–1.06) | 1.01 (0.96–1.07) | 1.01 (0.96–1.07) | 1.02 (0.96–1.08) |
| Sex | | | | |
| Female | Reference | Reference | Reference | Reference |
| Male | 3.18 (1.41–7.16)* | 3.45 (1.44–8.23)* | 2.49 (1.03–6.05)* | 2.63 (1.01–6.83)* |
| Education level | | | | |
| Primary or less | Reference | Reference | | Reference |
| Secondary or higher | 4.09 (1.83–9.11)* | 4.37 (1.90–10.05)* | - | 7.70 (3.02–19.60)* |
| Marital status | | | | |
| Married or cohabitating | Reference | Reference | | Reference |
| Divorced | 1.88 (0.70–5.00)* | 4.11 (1.40–12.03)* | - | 2.85 (0.78–10.43) |
| Widowed | 0.68 (0.20–2.26) | 0.98 (0.28–3.38) | - | 1.21 (0.33–4.49) |
| Never married | 1.27 (0.38–4.22) | 2.50 (0.70–8.94) | - | 3.28 (0.83–12.91) |
| Clinic location | | | | |
| Private zonal referral hospital | Reference | - | Reference | Reference |
| Public regional hospital | 2.92 (1.16–7.37)* | - | 4.23 (1.36–13.2)* | 6.33 (1.93–20.76)* |
| Depressive symptoms | | | | |
| No to mild | Reference | - | Reference | Reference |
| Moderate to severe | 2.07 (0.70–6.08) | - | 4.11 (1.18–14.3)* | 6.35 (1.69–23.87)* |
| ART adherence, n (%) | | | | |
| Complete adherence | Reference | - | Reference | Reference |
| Incomplete adherence | 1.11 (0.26–4.75) | - | 1.41 (0.31–6.38) | 1.98 (0.41–9.59) |
| CD4 nadir, median (IQR), cells/mm$^3$ | 0.99 (0.99–1.00) | - | 1.00 (0.99–1.00) | 1.00 (0.99–1.00) |
| Time on ART, median (IQR) days | 0.99 (0.99–1.00) | - | 1.00 (0.99–1.00) | 1.00 (0.99–1.00) |

Abbreviations: HR, hazard ratio, aHR, adjusted hazard ratio, CI, confidence interval, ART, antiretroviral therapy.

*P<0.05.

Model 1 (sociodemographic model): adjusted for age, sex, education level, and marital status.

Model 2 (clinical model): adjusted for age, sex, clinic location, depression, ART adherence, CD4 nadir, and time on ART.

Model 3 (full model): includes all variables included in model 1 or model 2.

moderate to severe symptoms of depression were significant predictors of death. Overall, these results highlight the success of HIV treatment programs in northern Tanzania. Despite this, efforts are still needed to ensure the benefits of ART are realized by all PLWH.

Our cohort consisted predominantly of individuals aged 35 years or older and women. This finding mirrors the background age and sex distribution of PLWH in Tanzania [22]. Furthermore, the higher proportion of women in our cohort reflects reported differences in care seeking behaviors between men and women with HIV disease [23]. Despite evidence of advanced HIV disease prior to ART initiation, the mortality rate of this treatment-experienced cohort was lower in comparison to previous studies evaluating mortality among PLWH on ART [24]. The low mortality in our cohort was likely secondary to the high proportion of participants achieving virologic suppression. The proportion of virologic suppression in this cohort is consistent with that reported in national HIV surveillance data [22]. Overall, our results indicate that HIV treatment programs in SSA are capable of achieving successful long-term outcomes for PLWH despite prior advanced disease. In particular, the low attrition, incremental immune recovery, and high virologic suppression among this cohort are encouraging.

HIV programs across SSA have consistently reported a higher risk of mortality for men on ART in comparison to women [25]. While this mortality differences has been best described in the early period after ART initiation, one study found that men continued to have a higher rate of mortality up to twelve years after ART initiation [6]. These observed sex differences in mortality may exist because men are more likely to initiate ART at advanced HIV disease, to be non-adherent to ART, and to fall out of care in comparison to women [26]. While some observed sex differences in mortality may be explained by differences that exist in the background population [27], the existing evidence support the need for interventions to increase men's engagement through all aspects of the HIV care cascade.

We found that depression was a risk factor for mortality in our study. Depression is an important but neglected public health problem in SSA and the prevalence of depression among PLWH may be higher in comparison to the background population [28]. Among PLWH, depression has been found to influence behaviors such as ART adherence and engagement in care [29]. A previously published study from our cohort found a positive relationship between depressive symptoms and incomplete ART adherence and an inverse relationship between depressive symptoms and virologic suppression [16]. Given the threat depression poses to PLWH in SSA through its negative impact on HIV care engagement and possibly mortality, the WHO currently recommends the incorporation of depression screening and treatment in HIV care [30]. However, more studies are needed to determine effective models for the integration of mental health into HIV care in SSA.

Patients receiving care at the CTC in the regional referral hospital in comparison to the larger zonal referral hospital were found to have a higher risk of death. In comparison to the zonal referral hospital, which is one of four consultant tertiary hospitals in Tanzania, the regional referral hospital in our study likely has fewer health and laboratory services as well as fewer service providers [31]. While the observed mortality difference between these two facilities may be confounded by another factor such as differences in socioeconomic status (SES) of the treated populations, studies have found that facility-level characteristics of HIV programs such as availability of supplies, turnaround time of lab results, staffing characteristics, and hours of operation can influence behaviors such as entry and engagement in care among PLWH [32, 33]. These findings highlight the complex and multidimensional factors that may impact outcomes for PLWH as they progress through the HIV care continuum. Efforts to improve outcomes for PLWH should reach beyond individual-level factors and include quality improvement measures at the facility level.

In our study, attainment of secondary education or higher was a predictor of mortality. Educational attainment is a frequently used and easily measured indicator of socioeconomic status (SES)

in many epidemiologic studies [34]. In SSA, studies have found that individuals with low SES continue to bear a high burden of HIV mortality despite the scale-up of ART [35, 36]. Given this association between SES and mortality, the reasons for the association between higher education attainment and mortality in our study are unclear. However, there are limitations in the use of education as a single measure of SES. Notably, education level may not capture the volatility of SES in later adulthood and may better represent childhood economic circumstances [37]. Furthermore, measurements of education level do not capture the quality of the educational experience, which likely plays a major role in socioeconomic growth. These results indicate that a thorough investigation of the association between SES and HIV outcomes is needed in our setting.

We adjudicated that 58% of deaths were HIV/AIDS-related using a verbal autopsy format. These deaths were not driven by early mortality as we observed no association between mortality and time on ART. Because most HIV/AIDS-related deaths are driven by infections [38], improved screening, prophylaxis, and treatment of opportunistic infections throughout the course of antiretroviral therapy could likely avert a number of deaths. As the HIV epidemic matures in our setting, we suspect that non-HIV/AIDS related deaths are likely to increase. Thus, monitoring causes of death could have significant public health value as such knowledge could inform regional and national guidelines.

This study has several limitations. First, our multivariable models violates the rule of thumb that proportional hazard models should contain a minimum of ten events per predictor variable (EPV) [39, 40]. However, models containing fewer than 10 EPV should not be discounted particularly in the setting of highly significant and plausible associations [41]. Second, all participants were enrolled from the CTCs of two hospitals in northern Tanzania. Thus, outcomes in this study population may not be generalizable to other regions in Tanzania or to other countries in SSA. Third, this study has survival bias in that those individuals who have survived and are maintained in care may not reflect the initial population initiated on ART. Thus, the results should be interpreted with the recognition that this cohort does not represent all patients started on ART, but rather those who have been engaged in care for some time. Fourth, verbal autopsy has several limitations. Notably, certainty is hindered by the quality of the medical records and recall bias among family and friends of the deceased [42]. Lastly, our analysis did not account for the influence of relevant noncommunicable diseases or their risk factors on mortality. With increasing age along with virologic suppression and immunologic recovery, noncommunicable diseases and their risk factors may play a role in mortality within this cohort.

In summary, the majority of this cohort demonstrated high levels of virologic suppression and immunologic recovery. Nonetheless, accumulating epidemiologic evidence, including the results of this study, indicate that not all PLWH realize the benefits of ART. HIV programs in SSA, which have been particularly effective in reducing the impact of the epidemic on women and children, must recognize the health impacts of under-reaching men and consider sex specific interventions to ensure equitable access to HIV care for all PLWH. In addition, the integration of mental health screening and treatment into HIV care could assure better quality of life and outcomes for PLWH in SSA. Lastly, a growing body of evidence suggest that behaviors and outcomes among PLWH in SSA are influenced by a complex array of factors that may include structural level challenges such as inequities among treatment facilities. Multi-level interventions will be required to maximize the ongoing benefit of ART for PLWH.

## Acknowledgments

We would like to thank the research participants, and members of the CHAT Research Team: Bernard Agala, Beatrice Lema, Yombya Madukwa, Restituta Mvungi, Amiri Mrema, Wendy Ricky, Ludovic Samora, and Blandina Zenze.

## Author Contributions

**Conceptualization:** Elizabeth A. Reddy, Jan Ostermann, Kathryn Whetten, Nathan M. Thielman.

**Data curation:** Jia Yao, Elizabeth A. Reddy.

**Formal analysis:** Deng B. Madut, Lawrence P. Park, Jia Yao, Jan Ostermann.

**Funding acquisition:** Kathryn Whetten, Nathan M. Thielman.

**Methodology:** Elizabeth A. Reddy, Nathan M. Thielman.

**Project administration:** Elizabeth A. Reddy, Kathryn Whetten.

**Writing – original draft:** Deng B. Madut, Lawrence P. Park, Jia Yao, Elizabeth A. Reddy, Bernard Njau, Jan Ostermann, Kathryn Whetten, Nathan M. Thielman.

**Writing – review & editing:** Deng B. Madut, Lawrence P. Park, Jia Yao, Elizabeth A. Reddy, Bernard Njau, Jan Ostermann, Kathryn Whetten, Nathan M. Thielman.

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
