## [Decision Letter · Decision Letter 0]

20 Apr 2020

PONE-D-19-29387

Predictors of mortality in treatment experienced HIV-infected patients in Northern Tanzania

PLOS ONE

Dear Dr. Madut,

Thank you for submitting your manuscript to PLOS ONE. After careful consideration, we feel that it has merit but does not fully meet PLOS ONE’s publication criteria as it currently stands. Therefore, we invite you to submit a revised version of the manuscript that addresses the points raised during the review process.

We would appreciate receiving your revised manuscript by Jun 04 2020 11:59PM. To enhance the reproducibility of your results, we recommend that if applicable you deposit your laboratory protocols in protocols.io, where a protocol can be assigned its own identifier (DOI) such that it can be cited independently in the future. For instructions see: http://journals.plos.org/plosone/s/submission-guidelines#loc-laboratory-protocols

We look forward to receiving your revised manuscript.

Kind regards,

Joel Msafiri Francis, MD, MS, PhD

Academic Editor

PLOS ONE

Reviewers' comments:

Reviewer's Responses to Questions

**Comments to the Author**

1. Is the manuscript technically sound, and do the data support the conclusions?

Reviewer #1: Partly

Reviewer #2: Yes

2. Has the statistical analysis been performed appropriately and rigorously? 

Reviewer #1: I Don't Know

Reviewer #2: Yes

3. Have the authors made all data underlying the findings in their manuscript fully available?

Reviewer #1: No

Reviewer #2: Yes

4. Is the manuscript presented in an intelligible fashion and written in standard English?

Reviewer #1: Yes

Reviewer #2: Yes

5. Review Comments to the Author

Reviewer #1: I am happy that the authors tried to tackle an important issue - the long term health of PLWH who are taking long term ARV in SSA - but these data do not add to what is already known for multiple reasons. The generalizability of the treatment experienced ART cohort is questionable. Depression was screened for, which is nice, but it does not appear that a mental health expert was very involved on the ground or in the analysis. The other slightly unique feature was repeated HIV viral load testing, but this was not considered in the mortality analysis - a missed opportunity. The methods overall are not inspiring and leave the conclusions prone to cohort analysis biases. See specific comments below.

Introduction:

• Need to better justify why early and late mortality factors may differ (i.e., justify the need for this paper)

o Wouldn’t late presentation to care cause long-lasting damage to the immune system?

o Are you suggesting that the ARVs or new risk factors (such as behavioral ones like alcohol, smoking) would be late but not early mortality RFs?

Methods

• Explain what ‘established infection receiving care’ means. Was there a minimum # years?

• A bit more details on the random point system – not familiar with it and how it can be implemented in Tanzania; what was the point of the sampling approach? To get a sample that represents the entire district? Or all of Tanzania? All of Africa?

• Prefer the term sex to gender, although am aware that transgender people would not self-identify in Tanzania’s current situation

• Was PHQ-9 done at every visit (every 6 months)?

• No mention of indications to start ART during the study.

• Why not use VL <1000 as evidence of adherence? Self-report is notoriously not good.

• No discussion of loss to follow-up or transfers out in the cohort and how it was mitigated, ascertained, etc.

• Not including the HIV-uninfected cohort (to comment on their mortality rates) seems like missed opportunity.

• How did you handle time varying predictors? Or were all predictors at the entry to cohort.

• Justify why you did these 3 models.

• No attempt made to adjust for the fact that patients who were retained for 700 days on ART and entered the study may be different from the overall clinic population and/or PLWH in the area. Nor was this major limitation well-acknowledged

• No mention of TB throughout this paper, the 1st cause of death in PLWH in SSA.

• They did not refer (or failed to mention it) depressed patients for mental health care, which raises some possible ethical concerns. Did you assess suicidal thoughts in persons reporting depression? Did you have a safety plan or mental health expert involved?

Results

• Many were excluded as not on ART; why and didn’t this introduce a strong bias into your analysis?

• These are very old data (from 2008); does it diminish current impact?

• Make it clear what the average time on ART was at censoring.

• Give more details on how mortality was determined. What was time from death to when the clinic became aware? Did you ever have inpatient records? How often was nothing other than outpatient file available? How often was verbal autopsy performed and with whom? How often did reviewers disagree, etc.

• Table 2- why not use HIV VL as adherence, since having repeated HIV VL is one of the unique elements of this cohort?

• Table 3 not very helpful

Discussion

• Must acknowledge the limitations of verbal autopsy

• Do you think most deaths are due to HIV/AIDS-related causes in a virally suppressed, high CD4 cohort? If so, why? Which HIV/AIDS-related causes of deaths are emerging or persisting here?

• If you wish to highlight depression in this cohort, why not demonstrate that depression led to reduced viral suppression and reduced CD4, which led to increased mortality. Isn’t that the mechanism. Or suicide – was it ascertained at all in depressed people

Reviewer #2: Title: Predictors of mortality in treatment experienced HIV-infected patients in Northern Tanzania

The manuscript presents a relevant and interesting topic. The manuscript has been written in clear and adequate details. However, authors may consider revisiting the following areas, for improving the manuscript.

1. Title page: A symbol (e.g. asterix) for corresponding author should be added on the corresponding author’s name. In addition, authors may consider adding key words.

2. Methods: Line 87, page 5 reads “Any patient 18 – 65 years old and residing in the districts of Moshi urban or Hai of the Kilimanjaro region…” It might be enlightening to explain the justification for the upper age limit.

3. Methods – Measures: 1. Socio and demographic characteristics – Given the established role of socioeconomic levels as a determinant of mortality, authors may consider adding to this analysis variables such as employment status or estimate for monthly income which explain better socioeconomic levels. As authors have correctly pointed out in the discussion, ‘level of education’ could be a poor measure of socioeconomic levels and therefore leading to ‘unexpected’ results.

4. Similarly, given accumulating data pointing to the contribution of non communicable diseases (hypertension and diabetes) to mortality among HIV patients on long treatment to ART, authors may consider adding to this analysis, variables such as BMI, blood pressure, blood glucose, liver and kidney function test. Also, it is important as part of discussion for this manuscript to build a case for routine HIV care and treatment to include some of these measures as well as conducting periodic screening for hypertension and diabetes as a strategy for reducing mortality in HIV patients who have been on long term exposure to ART.

5. Methods – Analysis: Line 139, page 7 reads “In multivariate analysis, we constructed three models…” It was not made clear on the justification of this analysis plan. Authors may consider elaborating this.

6. Discussion: page 17 first paragraph reads “Patients receiving care at the CTC in regional referral hospital in comparison to the larger zonal referral hospital were found to have higher risk of death”. While I completely agree with the author’s arguments. Authors may consider adding to their argument a potential indirect influence of socioeconomic status particularly on the side of patients given that patients’ socio-economic level will determine whether or not they choose to be cared at a public facility.

6. PLOS authors have the option to publish the peer review history of their article (what does this mean?). If published, this will include your full peer review and any attached files.

Reviewer #1: Yes: Michael Vinikoor

Reviewer #2: No

---

## [Author Response · Author response to Decision Letter 0]

7 Jun 2020

Response to reviewers letter attached.

---

## [Decision Letter · Decision Letter 1]

23 Jul 2020

PONE-D-19-29387R1

Predictors of mortality in treatment experienced HIV-infected patients in Northern Tanzania

PLOS ONE

Dear Dr. Madut,

Thank you for submitting your manuscript to PLOS ONE. After careful consideration, we feel that it has merit but does not fully meet PLOS ONE’s publication criteria as it currently stands. Therefore, we invite you to submit a revised version of the manuscript that addresses the points raised during the review process.

We look forward to receiving your revised manuscript.

Kind regards,

Joel Msafiri Francis, MD, MS, PhD

Academic Editor

PLOS ONE

Reviewers' comments:

Reviewer's Responses to Questions

**Comments to the Author**

1. If the authors have adequately addressed your comments raised in a previous round of review and you feel that this manuscript is now acceptable for publication, you may indicate that here to bypass the “Comments to the Author” section, enter your conflict of interest statement in the “Confidential to Editor” section, and submit your "Accept" recommendation.

Reviewer #1: (No Response)

Reviewer #2: All comments have been addressed

Reviewer #3: (No Response)

2. Is the manuscript technically sound, and do the data support the conclusions?

Reviewer #1: Partly

Reviewer #2: Yes

Reviewer #3: Yes

3. Has the statistical analysis been performed appropriately and rigorously? 

Reviewer #1: No

Reviewer #2: Yes

Reviewer #3: Yes

4. Have the authors made all data underlying the findings in their manuscript fully available?

Reviewer #1: Yes

Reviewer #2: Yes

Reviewer #3: Yes

5. Is the manuscript presented in an intelligible fashion and written in standard English?

Reviewer #1: Yes

Reviewer #2: Yes

Reviewer #3: Yes

6. Review Comments to the Author

Reviewer #1: This paper analyzes mortality predictors among persons with HIV in Northern Tanzania using data from 2008-2012, a period that one could considered the 'early ART era' there (with mostly D4T-based regimens), making the results less relevant to now. The analysis suffers from multiple issues that cannot be completely addressed although the authors have thoughtfully responded to my comments. The fact that the data were harmonized from multiple diverse cohorts and multiple forms of selection bias in the patient characteristics cannot really be addressed also reduce the impact of these results. Repeated measures of depression and inability to leverage an HIV-negative cohort (too demographically different) are missed opportunities. It is my opinion that this does not sufficiently build HIV-related knowledge to warrant publication in Plos One.

Reviewer #2: (No Response)

Reviewer #3: (No Response)

7. PLOS authors have the option to publish the peer review history of their article (what does this mean?). If published, this will include your full peer review and any attached files.

Reviewer #1: **Yes: **Michael Vinikoor

Reviewer #2: No

Reviewer #3: No

---

## [Author Response · Author response to Decision Letter 1]

21 Aug 2020

Response to the reviewers have been provided in the response to reviewer document.

---

## [Decision Letter · Decision Letter 2]

16 Sep 2020

PONE-D-19-29387R2

Predictors of mortality in treatment experienced HIV-infected patients in Northern Tanzania

PLOS ONE

Dear Dr. Madut,

Thank you for submitting your manuscript to PLOS ONE. After careful consideration, we feel that it has merit but does not fully meet PLOS ONE’s publication criteria as it currently stands. Therefore, we invite you to submit a revised version of the manuscript that addresses the points raised during the review process.

We look forward to receiving your revised manuscript.

Kind regards,

Joel Msafiri Francis, MD, MS, PhD

Academic Editor

PLOS ONE

Reviewers' comments:

Reviewer's Responses to Questions

**Comments to the Author**

1. If the authors have adequately addressed your comments raised in a previous round of review and you feel that this manuscript is now acceptable for publication, you may indicate that here to bypass the “Comments to the Author” section, enter your conflict of interest statement in the “Confidential to Editor” section, and submit your "Accept" recommendation.

Reviewer #2: (No Response)

Reviewer #3: All comments have been addressed

2. Is the manuscript technically sound, and do the data support the conclusions?

Reviewer #2: Yes

Reviewer #3: Yes

3. Has the statistical analysis been performed appropriately and rigorously? 

Reviewer #2: Yes

Reviewer #3: Yes

4. Have the authors made all data underlying the findings in their manuscript fully available?

Reviewer #2: Yes

Reviewer #3: Yes

5. Is the manuscript presented in an intelligible fashion and written in standard English?

Reviewer #2: Yes

Reviewer #3: Yes

6. Review Comments to the Author

Reviewer #2: For a manuscript on mortality among HIV infected patients on ART, it would be unfair to pass without a mention of high Body Mass Index, hypertension and type 2 diabetes mellitus ( high Blood Glucose) as potential risk factors to mortality. If such information were not collected, authors should strongly consider to include these as part of their study limitations.

Reviewer #3: All the comments have been addressed to my satisfaction. The manuscript is readability and data presentation have substantially improved.

7. PLOS authors have the option to publish the peer review history of their article (what does this mean?). If published, this will include your full peer review and any attached files.

Reviewer #2: No

Reviewer #3: No

---

## [Author Response · Author response to Decision Letter 2]

18 Sep 2020

Please see attached response to reviewers.

---

## [Editor Report · Decision Letter 3]

24 Sep 2020

Predictors of mortality in treatment experienced HIV-infected patients in Northern Tanzania

PONE-D-19-29387R3

Dear Dr. Madut,

We’re pleased to inform you that your manuscript has been judged scientifically suitable for publication and will be formally accepted for publication once it meets all outstanding technical requirements.

Kind regards,

Joel Msafiri Francis, MD, MS, PhD

Academic Editor

PLOS ONE
---

## [Editor Report · Acceptance letter]

28 Sep 2020

PONE-D-19-29387R3 

Predictors of mortality in treatment experienced HIV-infected patients in Northern Tanzania 

Dear Dr. Madut:

I'm pleased to inform you that your manuscript has been deemed suitable for publication in PLOS ONE. Congratulations! Your manuscript is now with our production department. 

Kind regards, 

on behalf of

Dr. Joel Msafiri Francis 

Academic Editor

PLOS ONE